# Developing a Deep-Learning-Based Coronary Artery Disease Detection Technique Using Computer Tomography Images

**DOI:** 10.3390/diagnostics13071312

**Published:** 2023-03-31

**Authors:** Abdul Rahaman Wahab Sait, Ashit Kumar Dutta

**Affiliations:** 1Department of Documents and Archive, Center of Documents and Administrative Communication, King Faisal University, P.O. Box 400, Hofuf 31982, Al-Ahsa, Saudi Arabia; 2Department of Computer Science and Information Systems, College of Applied Sciences, AlMaarefa University, Riyadh 13713, Saudi Arabia

**Keywords:** coronary artery disease, UNet++, cardiac arrests, convolutional neural networks, hyperparameter tuning

## Abstract

Coronary artery disease (CAD) is one of the major causes of fatalities across the globe. The recent developments in convolutional neural networks (CNN) allow researchers to detect CAD from computed tomography (CT) images. The CAD detection model assists physicians in identifying cardiac disease at earlier stages. The recent CAD detection models demand a high computational cost and a more significant number of images. Therefore, this study intends to develop a CNN-based CAD detection model. The researchers apply an image enhancement technique to improve the CT image quality. The authors employed You look only once (YOLO) V7 for extracting the features. Aquila optimization is used for optimizing the hyperparameters of the UNet++ model to predict CAD. The proposed feature extraction technique and hyperparameter tuning approach reduces the computational costs and improves the performance of the UNet++ model. Two datasets are utilized for evaluating the performance of the proposed CAD detection model. The experimental outcomes suggest that the proposed method achieves an accuracy, recall, precision, F1-score, Matthews correlation coefficient, and Kappa of 99.4, 98.5, 98.65, 98.6, 95.35, and 95 and 99.5, 98.95, 98.95, 98.95, 96.35, and 96.25 for datasets 1 and 2, respectively. In addition, the proposed model outperforms the recent techniques by obtaining the area under the receiver operating characteristic and precision-recall curve of 0.97 and 0.95, and 0.96 and 0.94 for datasets 1 and 2, respectively. Moreover, the proposed model obtained a better confidence interval and standard deviation of [98.64–98.72] and 0.0014, and [97.41–97.49] and 0.0019 for datasets 1 and 2, respectively. The study’s findings suggest that the proposed model can support physicians in identifying CAD with limited resources.

## 1. Introduction

Across the globe, cardiovascular diseases (CVD) are the leading cause of mortality, which accounts for an estimated 17.9 million deaths annually [1]. The most prevalent form of CVD is coronary artery disease (CAD), which frequently results in cardiac arrest. Coronary artery blockage leads to heart failure [2,3,4,5,6,7]. The heart relies on blood flow from the coronary arteries [8]. In developing countries, heart disease diagnosis and treatment are difficult due to the limited number of medical resources and professionals [9]. In order to avoid further damage to the patient, there is a demand for practical diagnostic tools and techniques. Both economically developed and underdeveloped nations are experiencing significant surges in the number of deaths from CVD [10]. Early CAD identification can save lives and lower healthcare costs [11,12,13,14,15,16]. Developing a reliable and non-invasive approach for early CAD identification is desirable. During the past few years, practitioners have significantly increased their utilization of computer technology to make decisions [17].

Physicians utilize conventional invasive methods to diagnose heart disease based on a patient’s medical history, physical tests, and symptoms [18]. Angiography is one of the most precise approaches for analyzing heart issues using conventional methods. However, it has a few limitations, such as a high cost, multiple side effects, and the requirement for extensive technological expertise [19]. Due to human error, conventional approaches frequently result in inaccurate diagnoses and additional delays. The coronary artery assessment through computed tomography (CT) is called coronary CT angiography (CCTA). A high-speed CT scan is performed on the cardiovascular system by administering a contrast agent through the intravenous route to the patient [20,21]. CCTA is used to identify atherosclerotic disease and evaluate abnormalities in the heart or blood vessels [22].

Machine learning (ML) is rapidly emerging as a game-changing tool for improving patient diagnoses in the healthcare sector [17]. It is an analytical method for huge and challenging programming tasks, including information transformation from medical records, pandemic forecasting, and genetic data analysis. Several studies suggest multiple approaches for identifying cardiac issues using machine learning [23,24,25,26]. The ML approach consists of several processes, including image preprocessing, feature extraction, training and parameter tuning, evaluating the model, and subsequently making predictions using the model. The classifier’s performance is based on the feature selection process. Several metrics have been described in the recent literature [27] for the evaluation of the ML-based model. These metrics include accuracy, sensitivity, specificity, and F1-score. Healthcare practitioners are primarily concerned about the ML-based model’s reliability and performance [28]. In addition, simplicity, interpretability, and computational complexity are essential criteria for implementing the CAD detection model in healthcare centers [29].

Deep learning (DL) is a relatively new ML technique with great promise for various classification problems [30]. DL offers a practical approach to building an end-to-end model using the raw medical image to predict a crucial disease [31]. In particular, the CNN model outperforms other methods in several image categorization problems. CNN identifies the key features and classifies images [32]. However, image annotation is one of the critical phases in medical image classification. High dataset dimensionality is a crucial issue for ML approaches [33]. The algorithm’s performance can be improved by weighting features, which reduce redundant data and prevent overfitting [33,34,35,36,37]. 

Alothman A. F. et al. [4] developed a CAD detection model using the DL technique. They employed the CNN model for classifying the CCTA images. In [7], the authors contributed to developing an automated classifier for patients with congestive heart failure. This classifier differentiates between individuals with a low risk and those with a high risk of complications. In [9], the authors presented a deep neural network technique for categorizing electrocardiogram data. The authors of [10] developed a clinical decision support system to evaluate heart failure. The researchers examined the efficacy of several ML classifiers, including neural networks, support vector machines, fuzzy rule systems, and random forest.

The existing CAD detection models demand high computational costs and training time for producing a reasonable outcome. It requires valuable features to identify an image’s key pattern. The recent models face difficulties in overcoming underfitting and overfitting issues. In addition, an effective feature extraction technique and hyperparameter-tuned CNN model can address the shortcomings of the existing CAD detection models. Therefore, this study intends to develop a CAD detection model using a CNN technique. The contributions of the study are:An image enhancement technique to improve the quality of the CT images.An intelligent feature extraction approach for extracting key features.A hyperparameter-tuned CNN technique for identifying CAD.

The remaining part of the paper is organized as follows: Section 2 presents the methodology of the proposed study. It highlights the research phases, dataset characteristics, and hyperparameter-tuning process. Section 3 outlines the experimental results on CCTA datasets. Section 4 discusses the study’s contribution and limitations. Finally, Section 5 concludes the study with its future directions.

## 2. Materials and Methods

The proposed CAD detection model uses the CNN technique for identifying CAD from the CT images. Figure 1 highlights the proposed CAD detection model. It contains image enhancement, feature extraction, and hyperparameter-tuned UNet++ models for predicting CAD using CCTA images. 

### 2.1. Dataset Characteristics

A total of two datasets are employed to train the models. Dataset 1 is publicly available in the repository [5]. The CCTA images of 500 patients are stored in the dataset. The images are classified into normal (50%) and abnormal (50%). The image is represented in 18 multiple views of a straightened coronary artery. The images are divided into training, validation, and test images. The authors have included 2364 images to balance the dataset.

The 3D CCTA images of 1000 patients are deposited in dataset 2. The images were captured using a Siemens 128-slice dual-source scanner. The size of the images is 512 × 512 × (206–275) voxels. The images were collected from the Guangdong Provincial People’s hospital between April 2012 and December 2018. The average ages of females and males were 59.98 and 57.68 years, respectively. The dataset repository [6] is publicly available for the researchers. In addition, it offers an image segmentation method for extracting images of coronary arteries from raw 3D images. Figure 2a,b are the raw images of datasets 1 and 2, respectively. Table 1 presents the characteristics of the dataset.

### 2.2. Proposed Methodology

Figure 3 highlights the research phases of the study. Phase 1 outlines the image preprocessing and feature extraction processes. Phase 2 describes the processes for classifying the CCTA images into CAD and No CAD. In this phase, the Aquila optimization (AO) algorithm [21] is employed for tuning the hyperparameters of the UNet++ model. Lastly, phase 3 presents the performance evaluation of the proposed model.

#### 2.2.1. Feature Extraction

In phase 1, the researchers follow the methods of [18] to enhance the image quality. A fuzzy function processes the standard CCTA image in the raster format. A discrete space is used to represent the height and width of an image. A mapping function maps the fuzzy image and the discrete space. The spatial information of the fuzzy image is located using a neighborhood function. The researchers modified the membership function of [18] to increase the pixel value. The membership function includes a rescaling function to enable the YOLO V7 model to rescale the images during feature extraction. Equation (1) shows the fuzzification process.
(1)$$\mathrm{Fuzzy}(\mathrm{CCTA}\;\mathrm{Image})=Int_{H.w}\left(CCTA\;image\right)+Mem_{H,w}\left(CCTA\;Image\right)$$
where $Int_{H.w}$ and $Mem_{H,w}$ are intensity and membership functions, and H and W are the height and width of the CCTA image. The defuzzification function applies the maxima for generating the enhanced CCTA image. Using the enhanced image, the researchers transform the images into different sizes and supply them to the subsequent phases. 

The images in dataset 2 are represented in 3D form, whereas the images of dataset 1 are expressed as the standard straightened arteries. To generate the straightened arteries from the 3D CCTA images, the researchers apply the centerline extraction [19] using the YOLO V7 model [20]. The YOLO V7 model identifies the centerlines using the anchor point between the coronary ostia and cardiac chambers. The arterial characteristics are generated using the central lines and area around the coronary vessels. In the subsequent steps, YOLO V7 extracts the features, which are forwarded to the CAD detection model.

#### 2.2.2. Fine-Tuned CNN Model

In phase 2, the author applies the AO algorithm and the UNet++ model to generate the outcome. CCTA image features are convolutionally processed using a linear filter and merged with a bias term. Then, the resulting feature map is passed through a non-linear activation function. Hence, each neuron gains input from an N × N area of a subset of feature maps of the prior or input layer. This neuron’s receptive fields comprise the combined regions of its receptive fields. As the same filter in the convolutional layer is used to probe all tolerable receptive fields of prior feature maps, the weights of neurons in the same feature map are always the same. 

During the training phase, the system acquires the shared weights, which may also be filters or kernels. The activation function is a mathematical equation for determining the outcome of a neural network [20]. The process is linked to each neuron of the network. The active neuron is used to support the model to make a prediction. The activation function determines the outcome of a neuron. The pooling layer triggers the non-linear function. This layer is assigned to reduce the number of values in the feature maps by identifying the important values of the previous convolutional layer. The dropout technique includes an additional hyperparameter and dropout rate, influencing the chance of removing or keeping layer outputs.

With UNet++, decoders from different U-Nets are densely coupled at the exact resolution [21]. As a result of structural improvements, UNet++ offers the following benefits. First, UNet++ embeds U-Nets of various depths in its design. The encoding and decoding processes of these U-Nets are interconnected, and the encoders are partially shared. All the individual UNets are trained in parallel with a standard image representation assistance by training UNet++ under deep supervision. This architecture enhances the total segmentation performance, and model pruning is made possible during the inference phase. In addition, the encoder and decoder of the UNet++ model allow the feature maps to be fused at a similar rate. The aggregation layer can determine how to merge feature maps transported via skip connections with decoder feature maps using UNet++’s new skip connections. The following section discusses the number of layers and the outcome of the training phase. In order to tune the hyperparameters of the UNet++ model, the researchers employ the specific features of the AO algorithm. Let P be the set of hyperparameters and consider a population of candidate solutions with the upper bound (U) and lower bound (L). In each iteration, an optimal solution is attained. Equations (2) and (3) present the candidate and random solutions for P.
(2)$$P=\left[\begin{array}{ccccc} P_{1,1} & \ldots & P_{1,j} & P_{1,Dim-1} & P_{1,Dim}\\ P_{2,1} & \ldots & P_{2,j} & \ldots & P_{2,Dim}\\ : & : & : & : & :\\ P_{N,1} & \ldots & P_{N,j} & P_{N,Dim-1} & P_{2,Dim} \end{array}\right]$$
where P represents the hyperparameters, N is the total number of parameters, and Dim is the dataset size.
(3)$$P_{i,j}=rand*\left(U_{j}-L_{j}\right)+L_{j}\;\;\;\;i=1,2,\ldots N;\;j=1,2,\ldots ,\mathrm{Dim}$$
where rand is the function to generate an anchor point for searching the parameter, i and j are the total number of parameters of the UNet++ model and the dataset’s size. The researchers derive narrowed exploration and exploitation features of the AO algorithm for finding the suitable hyperparameters of the UNet++ model. The AO agent considers the locations of hyperparameters as a prey area from a high soar and narrowly explores it using Equations (4) and (5).
(4)$$M_{1}\left(t+1\right)=M_{1}{}_{best}\left(t\right)\;X\;Levy\;\left(s\right)+M_{1}{}_{R}\left(t\right)+\left(Y-M_{1}\right)$$
where $M_{1}\left(t+1\right),\;M_{1}{}_{best},\;\;\mathrm{and}$ $M_{1}{}_{R}$ are the generative outcome at each iteration(t), s is the space, Y is the random location of the search space, and Levy(s) is a flight distribution function presented in Equation (5).
(5)$$Levy\left(s\right)=\frac{c*n*\sigma }{{\left|m\right|}^{\frac{1}{\beta }}}$$
where c, n, m, σ, and β are the constants for finding the hyperparameters.

Furthermore, narrow exploitation searches the hyperparameter using stochastic movements. Equation (6) shows the mathematical expression for the narrow exploitation.
(6)$$\begin{array}{c} M_{2}\left(t+1\right)=Q*\;M_{2}{}_{best}\left(t\right)-\left(G_{1}*M_{2}\left(t\right)*rand\right)\\ \;\;\;\;\;\;\;\;\;\;\;\;-\left(G_{2}*Levy\left(s\right)*rand\right)+G_{1} \end{array}$$
where $M_{2}\left(t+1\right)$ is the generative solution at iteration (t), Q represents the quality function, and $G_{1}\;\mathrm{and}\;G_{2}$ are movements of the AO agent. The researchers modified the quality function according to the UNet++ model’s performance.

#### 2.2.3. Performance Evaluation

Finally, the third phase evaluates the proposed method using the evaluation metrics, including accuracy, precision, recall, F1-score, Matthews correlation coefficient (MCC), and Kappa. The datasets are divided into a train set (70%) and a test set (30%). The number of parameters, learning rate, and testing time are computed for each model. The researchers compute the area under the receiver operating characteristic (AU-ROC) and the precision-recall (PR) curve for each CAD detection model. In addition, the confidence interval (CI) and the standard deviation (SD) are calculated to find the outcome’s uncertainty levels.

## 3. Results

To evaluate the performance of the proposed model, the researchers implemented the model in Windows 10 professional with an i7 processor, NVIDIA GeForce RTX 3060 Ti, and 8 GB RAM. Python 3.9, Keras, and Tensorflow libraries are used for constructing the proposed model. Yolo V7 [20] and UNet++ [21] are employed for developing the proposed model. In addition, the Alothman A.F. et al. model [4], Papandrianos N et al. model [7], Moon, J.H. et al. model [8], and Banerjee, R. et al. model [9] are used for performance comparison. The researcher trains the UNet++ model using datasets 1 and 2 under the AO environment. During the process, the proposed model scores a superior outcome at the 36th epoch and around the 34th epoch for datasets 1 and 2, respectively. The dropout ratios of 0.3 and 0.4 are used for datasets 1 and 2. These are used to address overfitting and underfitting issues. Finally, six layers, including two dropout layers, three fully connected layers, and a softmax layer, are integrated with the UNet++ model.

Table 2 presents the performance analysis of the proposed model on dataset 1. It indicates that the proposed model achieves an average accuracy and F1-measure of 98.85 and 98.37 during the training phase. In contrast, in the testing phase, it obtains a superior accuracy and F1-measure of 99.40 and 98.60. 

Table 3 reflects the proposed model performance on dataset 2. It is evident that the image enhancement and feature extraction processes support the proposed model to detect normal and abnormal CCTA images with optimal accuracy and F1-measure. 

Table 4 outlines the comparative analysis outcome of CAD using dataset 1. The proposed model outperforms the existing models by achieving accuracy, precision, recall, F1-measure, MCC, and Kappa of 99.40, 98.50, 98.65, 98.60, 95.35, and 95.00, respectively. 

Likewise, Table 5 displays the outcome of CAD detection models using dataset 2. The proposed model’s dropout and fully connected layers supported the UNet++ model to overcome the existing challenges of the CNN models in classifying the images. Thus, the performance of the proposed model is better compared to the baseline models. Figure 4 and Figure 5 highlight the performance of the CAD detection models on datasets 1 and 2, respectively. 

Figure 6 shows the AU-ROC and PR curves of the models using dataset 1. The proposed model learns the environment efficiently and handles the images effectively. In contrast, the current models face challenges in managing images of dataset 1. The proposed model obtained the AU-ROC and PR curve values of 0.97 and 0.95, which were higher than the baseline models of dataset 1.

Similarly, Figure 7 represents the AU-ROC and PR curve for dataset 2. Dataset 2 contains a smaller number of images compared to dataset 1. The recent models failed to generate a better AU-ROC and PR curve. In contrast, the proposed model generates the AU-ROC and PR curve values of 0.96 and 0.94, respectively. 

Table 6 highlights the computation cost of each model. The proposed model predicted the existence of CAD with fewer parameters in a shorter learning rate (1 × 10^−4^). In contrast, the Alothman A.F. et al. [4] model, Papandrianos N. et al. [7] model, Moon, J.H. et al. [8] model, and Banerjee R. et al. [9] model consumed a learning rate of 1 × 10^−4^, 1 × 10^−3^, 1 × 10^−3^, and 1 × 10^−3^, respectively. 

Table 7 reveals the CI and SD of the outcomes generated by the CAD detection models. The higher CI and SD values indicate that the proposed method’s results are highly reliable.

## 4. Discussion

Recently, there has been a demand for a lightweight CAD detection model for diagnosing patients at earlier stages. The CAD detection model helps the individual to recover from the illness. CCTA is one of the primary tools in detecting CAD. It offers a non-invasive evaluation of atherosclerotic plaque on the artery walls. The current CAD detection models require substantial computational resources and time. The researchers proposed a CAD detection model for classifying the CCTA images and identifying the existence of CAD.

Therefore, the researchers built a model using YOLO V7 and UNet++ models. The effectiveness of the model is evaluated using two datasets. Initially, the images are enhanced through a quality improvement process. Generally, the images are in grayscale with low quality. The proposed image enhancement increases the pixel size and removes the irrelevant objects from the primary images. Subsequently, YOLO V7 is applied to extract the CCTA images’ features. It is widely applied in object detection techniques. The researchers used this technique to identify the key features. Finally, the AO algorithm is used to tune the hyperparameters of the UNet++ model. The findings highlight that transfer learning can replace large datasets in potential AI-powered medical imaging to automate repetitive activities and prioritize unhealthy patients. The proposed method obtained an average accuracy of 99.40 and 99.50 for datasets 1 and 2, respectively. The outcome shows that the model correctly classifies CAD and No CAD from the CCTA images. The proposed feature extraction provided the critical features to the UNet++ model in making the decision. Precision, recall, and F1-measure values of 98.50, 98.65, and 98.60 for dataset 1 represent the effectiveness of the proposed model’s classification. The proposed model identified the relevant features of the straightened coronary arteries of dataset 1. In addition, the proposed model achieved superior precision, recall, and F1-measure values for dataset 2. The presented data preprocessing and feature extraction methods supplied the crucial features of straightened arteries to the UNet++ model. MCC and Kappa values of 96.35 and 96.25 highlight the binary classification ability of the suggested CAD detection model. Figure 6 and Figure 7 outline the AU-ROC and PR curve of the CAD detection models. It indicates the effectiveness of the proposed CAD detection model’s capability to handle true positive and false positive objects. 

However, the CNN model can produce a poor outcome due to the generalization ability. Thus, annotating or labeling the images is necessary to improve the performance of the YOLO V7 model. Transfer learning prevents overfitting and allows the generalization of tasks for other domains. It supports the UNet++ model to adjust the final weights concerning the features. The advantages of transfer learning using image embeddings with a feature extraction technique generate the highest average AUROC of 0.97 and 0.96 for datasets 1 and 2, respectively. The time necessary to train the proposed model was a few minutes, eliminating the requirement for a significant amount of computing resources and extensive training timeframes. The researcher achieves the study’s goal with limited resources by employing the CNN model. CAD detection models have demonstrated strong visual analysis, comprehension, and classification performance. The proposed model gradually reduces the input size, extracting features in parallel using convolutional layers. Images can be embedded to represent the input in a lower-dimensional environment properly. The fuzzy function offers an opportunity to improve the quality of images in the datasets. Improving the grayscale images enables the YOLO V7 model to identify valuable features. 

Furthermore, narrowed exploration and exploitation of the AO algorithm have identified the optimal set of hyperparameters for the UNet++ model. Although the UNet++ model contains an array of Unet models, it does not sufficiently address the overfitting issues. However, the hyperparameter optimization integrated a set of dropouts and fully connected layers with the UNet++ model. Thus, the proposed model achieves the study’s objective by developing a CAD detection model. The findings reveal that the proposed CAD detection model can help healthcare centers to identify CAD using limited computing resources. The CI and SD outcomes show that the results are reliable. The following outcomes of the comparative analysis reveal the proposed model’s significance in detecting CAD.

Alothman A.F. et al. [4] suggested a feature extraction strategy and a CNN model to identify CAD in the shortest amount of time while maintaining the highest level of accuracy. The effectiveness of the suggested model is examined using two datasets. The experimental results for the benchmark datasets reveal that the model achieved a better outcome with limited resources. However, the proposed model outperforms the model by producing a superior outcome. Papandrianos et al. [7] developed a model for detecting CAD using single-photon emission CT images. They applied an RGB-based CNN model for CAD detection. The model achieved an AUC score of 0.936. However, the proposed model obtained an AUC score of 0.97 and 0.96 on datasets 1 and 2. In addition, it produces a better outcome on grayscale CCTA images. 

Likewise, Moon J.H. et al. [8] proposed a DL model to detect CAD from 452 proper coronary artery angiography movie clips. In line with [8], the proposed model employs the YOLO V7 technique, which can be used for video clips. Moreover, the proposed model outperforms the Moon J.H. et al. model with limited resources. Table 6 outlines the computational complexities of the CAD detection models. It is evident that the proposed CAD detection model generated results with a few sets of parameters and a lower learning rate. Banerjee et al. [9] found a CNN long short-term memory approach for detecting CAD from the electrocardiogram images. Table 4 and Table 5 show that the Bannerjee et al. model produces low accuracy and F1-measure. The proposed model achieved a better outcome than the recent image classification [11,12,13,14,15,16,17,18]. The feature extraction technique supplied the practical features to support the proposed model and generate better insights from the CCTA images.

The proposed CAD detection generates an effective outcome on imbalanced datasets. However, there is a demand for future studies to overcome a few limitations of the proposed model. The multiple layers of the CNN model may require an additional training period. The UNet++ architecture requires an extensive search due to the varying depths. In an imbalanced dataset, the skip connection process may impose a restrictive fusion scheme to simultaneously force sub-networks to aggregate the feature maps.

## 5. Conclusions

The authors proposed a CAD detection model using the computed tomography images in this study. They intended to improve the performance of the CAD detection model using the effective feature extraction approach. The recent models require high computational costs to generate the outcome. Therefore, the authors proposed a three-phase method for detecting CAD from the images. In the first phase, an image enhancement technique using a fuzzy function improves an image’s quality. In addition, the authors applied the YOLO V7 technique to extract critical features. They improved the pixel value of the images to increase the YOLO V7 performance in extracting features from the grayscale images. The second phase used the AO algorithm for optimizing the hyperparameters of the UNet++ model with CCTA image datasets. The dropout layers are integrated with the model to address the overfitting issues. Finally, the third phase evaluated the performance of the proposed model. The state-of-the-art CAD detection models are compared with the proposed model. The comparative analysis revealed that the proposed model outperformed the recent CAD detection models. In addition, the computational cost required for the proposed model was lower than the others. The findings highlighted that the proposed model could support the healthcare center in developing countries to identify CAD in the initial stages. Moreover, the proposed model can be implemented with limited computational resources. However, future studies are required to minimize the training time and improve the performance of the CAD models with unbalanced data.

## Figures and Tables

**Figure 1 diagnostics-13-01312-f001:**
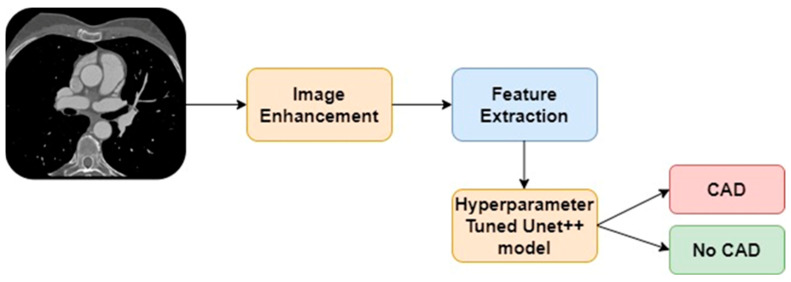
Proposed CAD detection model.

**Figure 2 diagnostics-13-01312-f002:**
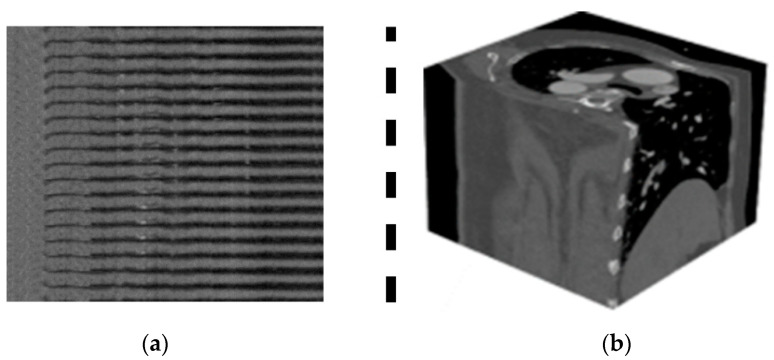
(**a**) Dataset 1. (**b**) Dataset 2.

**Figure 3 diagnostics-13-01312-f003:**
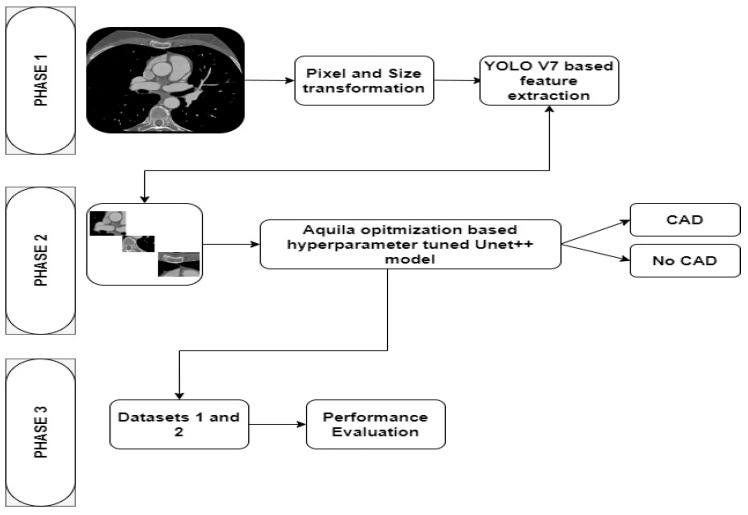
Research methodology.

**Figure 4 diagnostics-13-01312-f004:**
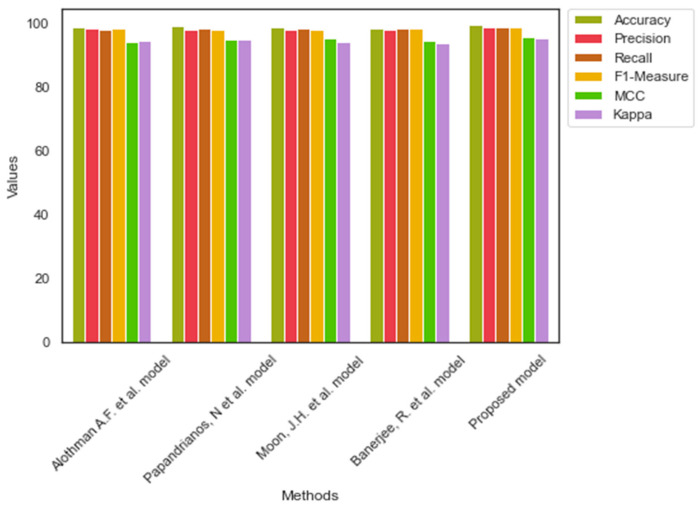
Comparative analysis of dataset 1 [4,7,8,9].

**Figure 5 diagnostics-13-01312-f005:**
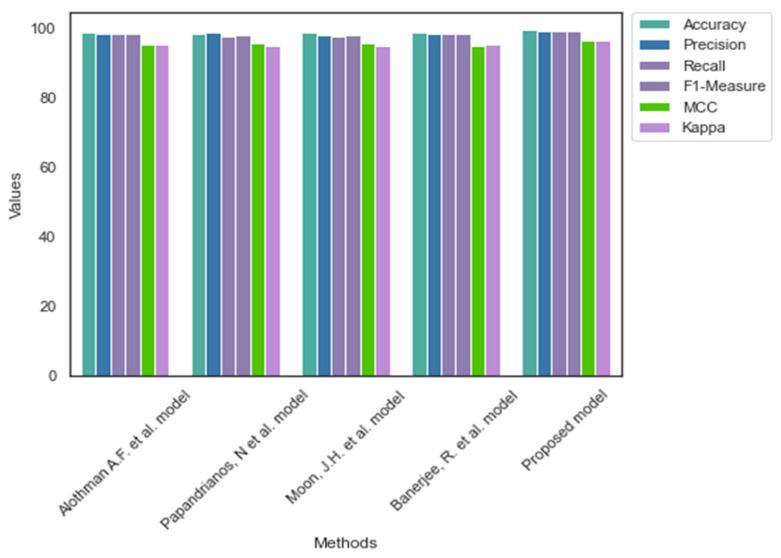
Comparative analysis of dataset 2 [4,7,8,9].

**Figure 6 diagnostics-13-01312-f006:**
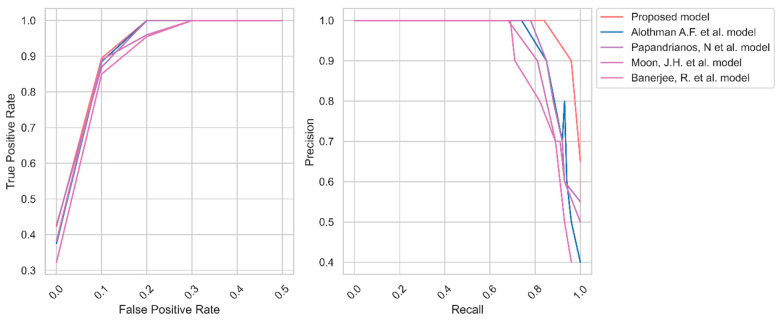
AU-ROC and PR curve of CAD for dataset 1 [4,7,8,9].

**Figure 7 diagnostics-13-01312-f007:**
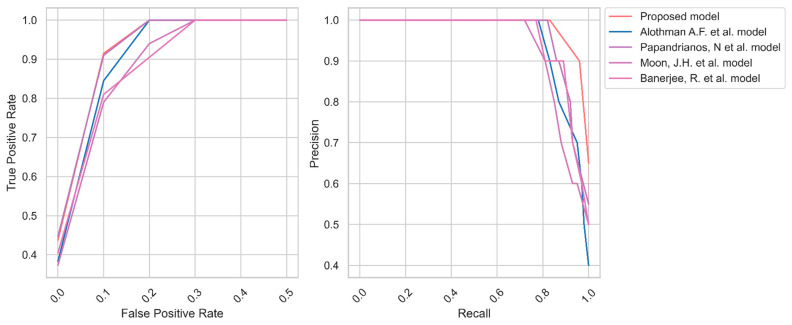
AU-ROC and PR curve of CAD for dataset 2 [4,7,8,9].

**Table 1 diagnostics-13-01312-t001:** Dataset characteristics.

Dataset	Number of Images	Number of Patients	CAD	No CAD	Classifications
Dataset 1	2364	500	1182	1182	2
Dataset 2	1000	1000	503	497	2

**Table 2 diagnostics-13-01312-t002:** Performance analysis for dataset 1.

Methods/Measures	Accuracy	Precision	Recall	F1-Measure	MCC	Kappa
Training						
CAD	98.60	98.10	98.60	98.35	95.30	95.60
No CAD	99.10	98.40	98.40	98.40	95.40	94.90
Average	98.85	98.25	98.50	98.37	95.35	95.25
Testing						
CAD	99.20	98.60	98.60	98.65	95.60	95.20
No CAD	99.60	98.40	98.70	98.55	95.10	94.80
Average	99.40	98.50	98.65	98.60	95.35	95.00

**Table 3 diagnostics-13-01312-t003:** Performance analysis for dataset 2.

Methods/Measures	Accuracy	Precision	Recall	F1-Measure	MCC	Kappa
Training						
CAD	98.70	98.80	98.70	98.75	96.40	96.30
No CAD	99.10	99.20	99.40	99.30	96.30	96.10
Average	98.90	99.00	99.05	99.02	96.35	96.20
Testing						
CAD	99.40	98.80	98.60	98.70	96.40	96.30
No CAD	99.60	99.10	99.30	99.20	96.30	96.20
Average	99.50	98.95	98.95	98.95	96.35	96.25

**Table 4 diagnostics-13-01312-t004:** Comparative analysis of CAD detection models using dataset 1.

Models/Measures	Accuracy	Precision	Recall	F1-Measure	MCC	Kappa
Alothman A.F. et al. [4] model	98.60	98.20	97.80	98.00	94.10	94.20
Papandrianos, N. et al. [7] model	98.90	97.80	98.10	97.95	94.80	94.60
Moon, J.H. et al. [8] model	98.50	97.60	98.20	97.90	95.10	93.80
Banerjee, R. et al. [9] model	98.20	97.80	98.30	98.05	94.30	93.70
Proposed model	99.40	98.50	98.65	98.60	95.35	95.00

**Table 5 diagnostics-13-01312-t005:** Comparative analysis of CAD detection models using dataset 2.

Models/Measures	Accuracy	Precision	Recall	F1-Measure	MCC	Kappa
Alothman A.F. et al. [4] model	98.60	98.20	98.10	98.15	95.30	95.10
Papandrianos, N. et al. [7] model	98.30	98.60	97.40	98.00	95.40	94.90
Moon, J.H. et al. [8] model	98.50	97.90	97.60	97.75	95.70	94.70
Banerjee, R. et al. [9] model	98.70	98.20	98.40	98.30	94.80	95.20
Proposed model	99.50	98.95	98.95	98.95	96.35	96.25

**Table 6 diagnostics-13-01312-t006:** Computational requirements for the CAD detection model.

Methods/Dataset	Dataset 1			Dataset 2		
	No. of Parameters	Learning Rate	Learning Time (seconds)	No. of Parameters	Learning Rate	Learning Time (seconds)
Alothman A.F. et al. [4] model	4.3 M	1 × 10^−4^	1.92	5.2 M	1 × 10^−3^	1.98
Papandrianos, N. et al. [7] model	11.2 M	1 × 10^−3^	2.1	6.3 M	1 × 10^−3^	2.45
Moon, J.H. et al. [8] model	7.4 M	1 × 10^−3^	2.36	11.2 M	1 × 10^−4^	2.27
Banerjee, R. et al. [9] model	14.6 M	1 × 10^−3^	2.3	6.1 M	1 × 10^−5^	2.3
Proposed model	3.6 M	1 × 10^−4^	1.4	3.7 M	1 × 10^−4^	1.5

**Table 7 diagnostics-13-01312-t007:** Uncertainty levels of the CAD detection model outcomes.

Methods/Dataset	Dataset 1		Dataset 2	
	CI	SD	CI	SD
Alothman A.F. et al. [4] model	[98.55–98.61]	0.0017	[96.62–96.71]	0.0021
Papandrianos, N. et al. [7] model	[97.41–97.48]	0.0021	[95.37–95.41]	0.0042
Moon, J.H. et al. [8] model	[97.32–97.42]	0.0016	[95.82–95.91]	0.0029
Banerjee, R. et al. [9] model	[97.91–98.02]	0.0019	[95.96–96.02]	0.0031
Proposed model	[98.64–98.72]	0.0014	[97.41–97.49]	0.0019

## Data Availability

The datasets can be found in the following repositories: https://data.mendeley.com/datasets/fk6rys63h9/1 (accessed on 27 February 2023) and https://github.com/XiaoweiXu/ImageCAS-A-Large-Scale-Dataset-and-Benchmark-for-Coronary-Artery-Segmentation-based-on-CT (accessed on 27 February 2023).

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
