# Peer review of "Developing a Deep-Learning-Based Coronary Artery Disease Detection Technique Using Computer Tomography Images"

_diagnostics, 2023, doi:10.3390/diagnostics13071312_

Round 1

Reviewer 1 Report

The authors present an article describing a deep learning-based approach for image classification to diagnose and classify Coronary Artery Disease.

The article is well written and the work is well done, I reserve the right to make a few observations:

1) There is one missing citation on line 133.

2) The formulas should be centred on the page 

3) Table 1, describes the dataset, and should show the number of samples in the two classes CAD and No CAD respectively

Author Response

The authors present an article describing a deep learning-based approach for image classification to diagnose and classify Coronary Artery Disease.

The article is well written and the work is well done, I reserve the right to make a few observations:

Response: Thank you very much for your comments.

1) There is one missing citation on line 133.

Response: The missing citation is included in line number 148.

2) The formulas should be centred on the page 

Response: As per the suggestion, the formulas are centered.

3) Table 1, describes the dataset, and should show the number of samples in the two classes CAD and No CAD respectively

Response: As per the suggestion, two columns are added in the table 1.

Reviewer 2 Report

The authors have developed a machine-learning model for detecting cardiovascular disease using the CNN technique. The model proposed by the authors solves a two-class classification problem by combining image enhancement preprocessing for input images using fuzzy functions, feature extraction using YOLO V7, and UNet++ with hyperparameters optimized by the Aquila algorithm.

The authors applied two open datasets and verified the performance of the proposed model with the model developed in previous research as a baseline.

About the entire manuscript:

I noticed some redundant descriptions, grammatical mistakes, and spelling errors. The authors should ask an appropriate proofreading institute to revise the manuscript.

A lot of mathematical formulas are used, but if the applied algorithm refers to previous research, rather than the theory derived by the authors, I think that it should not be shown unnecessarily, to prevent the manuscript from being difficult to understand.

The novelty of this paper seems to be combining several existing techniques, so I would suggest configuring the manuscript to make it clear.

The current version is very confusing.

About datasets:

The subject in the sentence is very difficult to understand, so please use passive voice to make it easier to understand. The used datasets do not seem to be general cross-sections of CT imaging. Please show sample images.

About preprocessing:

Although some equations are shown, they are very difficult for me to understand because of variables or flow of formula conversion were not explained. In addition, the reference number is missing, so it was not possible to confirm the correctness.

How about splitting the Method section into several minor sections for each algorithm description?

Does UNet++ output the results of two-class classification? Or are you creating an architecture different from the previous research on UNet++? If so, please explain its structure.

Please unify the significant digits of values in the Table.

Figure 3: A line graph is used to graphically represent changes in series data. Not only is this misused, but the results shown are perfectly understandable in the Table, and I don't think you need this graph at all.

Figure 4: If you want to show this result, please put the results for each model in a bar chart in one figure.

Table 6: I am not sure what you're referring to as computational complexities. Shouldn't the Learning time be shown instead of the Testing time?

In the discussion, I could not find any description of the meaning of the results or the grounds on which the results were obtained. The discussions here are only a list of results and facts. Also, no limitations are indicated, the scope covered by the study is not clear, and the rationality and validity of the asserted opinions are not guaranteed.

Author Response

The authors have developed a machine-learning model for detecting cardiovascular disease using the CNN technique. The model proposed by the authors solves a two-class classification problem by combining image enhancement preprocessing for input images using fuzzy functions, feature extraction using YOLO V7, and UNet++ with hyperparameters optimized by the Aquila algorithm.

The authors applied two open datasets and verified the performance of the proposed model with the model developed in previous research as a baseline. 

About the entire manuscript:

I noticed some redundant descriptions, grammatical mistakes, and spelling errors. The authors should ask an appropriate proofreading institute to revise the manuscript.

A lot of mathematical formulas are used, but if the applied algorithm refers to previous research, rather than the theory derived by the authors, I think that it should not be shown unnecessarily, to prevent the manuscript from being difficult to understand.

The novelty of this paper seems to be combining several existing techniques, so I would suggest configuring the manuscript to make it clear.

The current version is very confusing.

Response: Thank you for your suggestion. We have requested our colleague in the department of English, AlMaarefa University, Saudi Arabia for making proofread to rectify the grammatical flaws.We have removed the mathematical formulas and retained the derived expressions from the previous research papers.  

About datasets:

The subject in the sentence is very difficult to understand, so please use passive voice to make it easier to understand. The used datasets do not seem to be general cross-sections of CT imaging. Please show sample images.

Response: As per the suggestion, we have included the figures in line no. 131. 

About preprocessing:

Although some equations are shown, they are very difficult for me to understand because of variables or flow of formula conversion were not explained. In addition, the reference number is missing, so it was not possible to confirm the correctness.

 Response: The mathematical expressions were removed from the data processing section and simplified expression with reference number is added in line no. 148.

How about splitting the Method section into several minor sections for each algorithm description?

 Response: Thank you for the suggestion. We have made multiple sub sections in the section 2.

Does UNet++ output the results of two-class classification? Or are you creating an architecture different from the previous research on UNet++? If so, please explain its structure.

 Response: We employed the existing Unet++ model and fine tuned it for binary classification.

 Please unify the significant digits of values in the Table.

Response: Thank you for your suggestion. We have unified the values in the tables. 

Figure 3: A line graph is used to graphically represent changes in series data. Not only is this misused, but the results shown are perfectly understandable in the Table, and I don't think you need this graph at all.

Response: Thank you for your valuable inputs. We removed the figure 3.

Figure 4: If you want to show this result, please put the results for each model in a bar chart in one figure.

 Response: As per the suggestion, we have made figures 4 and 5 for each dataset and included in line number 275.

Table 6: I am not sure what you're referring to as computational complexities. Shouldn't the Learning time be shown instead of the Testing time?

 Response: Table 6 representing the computational requirements for making the prediction. We replaced the testing time by learning time.

 In the discussion, I could not find any description of the meaning of the results or the grounds on which the results were obtained. The discussions here are only a list of results and facts. Also, no limitations are indicated, the scope covered by the study is not clear, and the rationality and validity of the asserted opinions are not guaranteed.

Response: Thank you for the comments. We included the descriptions in line number 306, 322, and 331. The limitations are mentioned in the line number 385.

Round 2

Reviewer 2 Report

I would like to offer some minor comments.

1. The fonts of values in the tables are not consistent, is this intentional? In addition, I think it would be easier to interpret if the best value is bolded.

2. You probably don't need the statement on line 267. (There is a similar expression on line 270.)

3. Regarding the structure of the discussion section, I recommend that you write with an awareness of the connection between the sentences. In particular, there are many paragraph divisions, but it seems that there are some unnecessary line breaks. Please confirm.

4. I don't think there should be a section for Limitation (written as 3.1). Given the general structure of academic papers, it is expected that the discussion section would end with limitations or future issues explanation. Please confirm.

Author Response

I would like to offer some minor comments.

 Response: Thank you for your insights.

  1. The fonts of values in the tables are not consistent, is this intentional? In addition, I think it would be easier to interpret if the best value is bolded.

 Response: As per the suggestions, we have changes the font values of the tables and made the significance value in bold.

  1. You probably don't need the statement on line 267. (There is a similar expression on line 270.)

 Response: As per the comment, we removed the line 267.

  1. Regarding the structure of the discussion section, I recommend that you write with an awareness of the connection between the sentences. In particular, there are many paragraph divisions, but it seems that there are some unnecessary line breaks. Please confirm.

 Response: Thank you for your valuable insights. We removed the unnecessary line break and introduced a connection terms between the paragraphs.

  1. I don't think there should be a section for Limitation (written as 3.1). Given the general structure of academic papers, it is expected that the discussion section would end with limitations or future issues explanation. Please confirm.

Response: Thank you for your comments. We removed the 3.1 limitation from the discussion section.